# A flashing light may not be that flashy: A systematic review on critical fusion frequencies

**Alix Lafitte**[1,2]*, **Romain Sordello**[1], **Marc Legrand**[1,2,3], **Virginie Nicolas**[4,5], **Gaël Obein**[2,6], **Yorick Reyjol**[1]

1 PatriNat (Office Français de la Biodiversité (OFB), Muséum National d'Histoire Naturelle (MNHN), Centre National de la Recherche Scientifique (CNRS)), Paris, France, 2 Association Française de l'Eclairage (AFE), Paris, France, 3 Université Jean Monnet, Saint-Etienne, France, 4 Association des Concepteurs lumière et Eclairagistes (ACE), Paris, France, 5 Concepto, Arcueil, France, 6 Laboratoire National de métrologie et d'Essais—Conservatoire National des Arts et Métiers (LNE-CNAM), Saint-Denis, France

* alix.lafitte@mnhn.fr

**Data Availability Statement:** All relevant data are within the paper and its Supporting Information files.

**Funding:** This research was funded thanks to the support of the AFE (French Association on

## Abstract

### Background

Light pollution could represent one of the main drivers behind the current biodiversity erosion. While the effects of many light components on biodiversity have already been studied, the influence of flicker remains poorly understood. The determination of the threshold frequency at which a flickering light is perceived as continuous by a species, usually called the Critical Fusion Frequency (CFF), could thus help further identify the impacts of artificial lighting on animals.

### Objective

This review aimed at answering the following questions: what is the distribution of CFF between species? Are there differences in how flicker is perceived between taxonomic classes? Which species are more at risk of being impacted by artificial lighting flicker?

### Methods

Citations were extracted from three literature databases and were then screened successively on their titles, abstracts and full-texts. Included studies were critically appraised to assess their validity. All relevant data were extracted and analysed to determine the distribution of CFF in the animal kingdom and the influence of experimental designs and species traits on CFF.

### Results

At first, 4881 citations were found. Screening and critical appraisal provided 200 CFF values for 156 species. Reported values of CFF varied from a maximum of between 300 Hz and 500 Hz for the beetle *Melanophila acuminata* D. to a mean of 0.57 (± 0.08) Hz for the snail

Lighting), the ACE (French Association of Lighting Designers and Lighting Engineers), Citeos and PatriNat (French Office for Biodiversity (OFB) – French National Museum of Natural History (MNHN) – French National Centre for Scientific Research (CNRS)).

**Competing interests:** The authors have read the journal's policy and declare the following competing interests: Gaël Obein is the president of the AFE (French Lighting Association) and Virginie Nicolas is the president of the ACE (French Association of Lighting Designers and Lighting Engineers).

*Lissachatina fulica* B. Insects and birds had higher CFF than all other studied taxa. Irrespective of taxon, nocturnal species had lower CFF than diurnal and crepuscular ones.

## Conclusions

We identified nine crepuscular and nocturnal species that could be impacted by the potential adverse effects of anthropogenic light flicker. We emphasize that there remains a huge gap in our knowledge of flicker perception by animals, which could potentially be hampering our understanding of its impacts on biodiversity, especially in key taxa like bats, nocturnal birds and insects.

## Introduction

We are currently facing a major crisis of biodiversity erosion [1] with species becoming extinct more and more rapidly, notably since the 1990s [2]. Simultaneously, satellite-detectable light has increased by at least 49% between 1992 and 2017, notably due to urbanisation and Light-Emitting Diode (LED) 'rebound effect' [3]. Furthermore, light pollution, generated by anthropogenic light sources, has been identified as one of the main factors behind the current biodiversity erosion [4, 5].

The effects of artificial light at night (ALAN) on species and ecosystems are now more and more acknowledged [6, 7] and have been linked to alteration of physiology, behaviour, reproduction, habitat use, mobility and inter-species relationships [8]. In plants, light pollution can disrupt the phenology and advance the timing of budburst as shown by ffrench-Constant *et al.* (2016) [9]. At higher trophic levels, Owens *et al.* (2020) [10] reported that ALAN may alter insects' spatial distribution and life history, thus representing one of the main anthropogenic drivers behind insect decline. For birds, Dickerson, Hall & Jones (2022) [11] found that ALAN could suppress the nocturnal singing behaviour of the willie wagtail *Rhipidura leucophrys* L. and could be linked to a higher predation risk as individuals would be more easily spotted by predators under lighted conditions. Artificial light can also have adverse effects in aquatic environments and has been linked to reduced fitness in the common clownfish *Amphiprion ocellaris* C. [12]. In addition, ALAN could also have deleterious impacts on ecosystem functioning by, for example, disrupting two key ecosystem processes, namely pollination and seed dispersal [13, 14].

Impacts on biological organisms have been linked to several key components of artificial lighting, such as intensity [15–17], spectral composition [18–20], or temporality [21, 22]. Flicker frequency, another component of lighting [23], can be a consequence of either the alternating nature of power supply (i.e. 50 Hz in Europe and 60 Hz in the United States) or dynamic flashing rates of anthropogenic light sources and could represent a possible additional and significant source of impacts. For instance, Greenwood *et al.* (2004) [24] found a significant behavioural preference for a continuous light compared to a flickering one in the European starling, *Sturnus vulgaris* L. Sautter, Cocchi & Schenk (2008) [25] showed that a 1.5 Hz flickering light could increase the time that rats took to get out of a water labyrinth. Barroso *et al.* (2017) [26] examined the number of captured nocturnal insects between continuous and flickering light sources and found a potential flicker avoidance effect as lowered numbers of Diptera, Hemiptera and Lepidoptera were caught in traps associated with a flickering light. Flicker can also have potential significant effects on human health, as Wilkins *et al.* showed as early as in 1989 [27]. The qualification and quantification of the different impacts of flickering

lights is becoming increasingly important as new technologies such as LEDs now allow for more advanced dynamic lighting to appear, notably concerning shopfronts and ad panels, or the development of traffic-regulated street lamps [7, 23].

Species perception of flicker may depend on their temporal resolution, which can be defined as the ability to resolve rapid movements [28]. Each species is likely to have a different temporal resolution, shaped, among other parameters, by its ecology—e.g. foraging behaviour and habitat [29, 30]. A species temporal resolution can be estimated using Flicker Fusion Frequency (FFF), defined as the threshold frequency at which a flickering light is perceived as continuous [31]. Because FFF values can vary with light intensity, Critical Fusion Frequency (CFF) is defined as the maximum flicker fusion frequency at any light intensity—i.e. the point where any further increase of light intensity does not lead to an increase of FFF [32]. A first step before assessing the impacts of flashing artificial light on animals is to gain better knowledge of variations in CFF across the animal kingdom. To this end, Inger *et al.* (2014) [33] published a set of 93 CFF measurements for 81 species and observed that a substantial amount of animal taxa, and especially birds and insects, were likely to perceive the flicker of a lamp on a 50 Hz or 60 Hz power supply. Healy *et al.* (2013) [34] also compiled a sample of 34 specific CFF values and showed that body mass and metabolic rate were correlated to temporal resolution. Both reviews, while useful, are not based on specific methods of evidence synthesis. The comprehensiveness or the transparency of their search strategy, screening process (inclusion/exclusion decisions of all articles) or data extraction may thus be limited. For decision-makers to be informed in the best possible way and to reach highly beneficial biodiversity protective actions, we argue that a more comprehensive and transparent literature survey is therefore needed [35, 36].

Here, we present a systematic review of animals' critical fusion frequencies, using the method of systematic maps and reviews as recommended by the Collaboration for Environmental Evidence [37]. Systematic reviews are based on standardized protocols that have been developed in the field of ecology over the last few years and have proven very successful to provide strong scientific evidence for practitioners [38–40]. To consolidate the existing knowledge and earlier published reviews, we used a comprehensive search strategy based on several databases and performed a critical analysis of accepted studies in order to assess their susceptibility to bias and to attribute levels of confidence in CFF values. A database was produced, containing all collected CFF values, as well as their associated metadata. Our analysis identified patterns in the distribution of CFF across the animal kingdom, which may notably result from the influence of some species traits. This work focused on wild and domestic animals, excluding humans, and aimed at answering the following questions: what is the distribution of CFF between species? Are there differences in how flicker is perceived between taxonomic classes? Which species are more at risk of being impacted by artificial lighting flicker?

## Material & methods

This review, requested by professional lighting associations—i.e. the French Association on Lighting (AFE) and the French Association of Lighting Designers and Lighting Engineers (ACE)—followed the method of systemic review recommended by the Collaboration for Environmental Evidence (CEE) [37] and conformed to ROSES reporting standards [41] (S1 File). Deviations from CEE standards are listed in the section "Review limitations".

### Search for literature

We conducted a search for literature on three accessible databases from the Web of Science platform (Clarivate): Web of Science Core Collection (WOSCC), Biological Abstracts, and

Zoological Records—using the access rights provided by the French National Museum of Natural History (MNHN). These databases were selected because they cover both biology and ecology and because their functionalities enable an advanced search of literature. The WOSCC search included the following citation indexes: SCI–EXPANDED, SSCI, A&HCI, CPCI–S, CPCI–SSH, BKCI–S, BKCI–SSH, ESCI and CCR–EXPANDED. A search string was built with English terms as follows:

("fusion frequenc*"OR "flicker threshold$" OR "flicker sensitivit*" OR "flicker detection$" OR "flicker vision$" OR "flicker fusion" OR "flicker frequenc*" OR "temporal sinusoidally-modulated full-field luminance variation$")

This search string resulted from an iterative building process achieved by both ecological and physic experts from the MNHN and the National Conservatoire of Arts and Crafts (CNAM) and was used to reach the best comprehensiveness—i.e. the best recovery from a previously established test list of relevant articles. This test list included 60 articles, of which 56 came from Inger *et al.* (2014) [33] and 4 were included by the review team (S3 File). Of these 60 articles, 39 were found in the three databases that we used, providing an 85% comprehensiveness (33/39). The final search was conducted on "Topic" (TS) on 1 February 2021.

## Screening process

All citations were exported and screened through a three-stage process: firstly, on titles, then, on abstracts and finally, on full-texts. Each of the three screening stages was based on predefined inclusion/exclusion criteria—i.e. population, exposure and outcome—according to our review questions. Thence, we included all wild and domesticated animal species while excluding humans and other living organisms—e.g. plants, bacteria. Only artificial light sources at all wavelengths and colour temperatures were taken into account, whereas natural light sources or other types of waves like noise were excluded. We retained only citations measuring behavioural or physiological responses.

At the full-text screening stage, some additional criteria that could not be assessed at previous stages were included, regarding the language and the content and type of document. Only articles written in English and French were considered. We acknowledge that only including articles in those two languages constitutes a potential bias to our systematic review but this could not be avoided based on the linguistic competences of the review team. Articles which corresponded only to an abstract were excluded, as well as reviews and studies based strictly on modelling to keep solely studies with real *in-situ* or *ex-situ* collected data. At last, articles dealing with flicker which did not provide any CFF values were also discarded.

Each step of the screening process was performed by two reviewers (ML and RS), in accordance with CEE guidelines [37]. To assess the consistency of the inclusion/exclusion decisions between screeners, a Randolph's Kappa coefficient was computed on a random set of 5% of all articles to be screened at each step. The process was repeated until reaching a Kappa coefficient value higher than 0.6. In any case, all disagreements were discussed and resolved before beginning the screening process.

## Other sources of literature

To complete our search for literature strategy, we retrieved documents from other sources. First, we extracted references identified when carrying out critical appraisal and where CFF data were cited but were not present in our initial corpus [28, 42–45]. Additionally, we included values extracted from the two main previously published reviews on CFF in our corpus [33, 34]. Other relevant articles found by the review team, but not directly found in the three considered databases, were also added to the final corpus. As these articles all dealt with

CFF, they did not go through any screening stage and were directly included in our corpus. A call for literature—and in particular non peer-reviewed articles published in French or in English—was also carried out by sending emails to a group of 40 experts on 12 February 2021. The corresponding documents were screened on full-texts according to the same inclusion/exclusion criteria as described above.

## Critical appraisal

All the articles accepted on full-texts were split into studies—a study corresponding to one species and one method—in order to carry out a critical appraisal and assess their validity. First, a test was conducted on a subsample of studies by two reviewers (RS and AL) before critical appraisal was performed by AL for all studies. To define the criteria of this appraisal, a golden standard protocol was firstly determined in the context of an ideal study, supposedly granted with unlimited financing, time and workforce [37]. Six criteria were identified:

- the number of individuals (Replication criterion),

- the number of measures (Repetition criterion),

- the presence of a control (Control criterion); namely a reference electrode in the case of an electroretinogram or a continuous light for a behavioural experiment,

- the randomisation of individuals throughout experimental groups (Randomisation criterion),

- how specimens were handled before the experiment (Population criterion),

- how specimens were exposed to the treatment during the experiment (Exposition criterion).

Each criterion was assigned a 'high', 'medium' or 'low' risk of bias (see S6 File for details). Finally, an overall risk of bias was assigned for each accepted study:

- 'high' for a study with three high-risk-of-bias criteria,

- 'medium' for a study which had a medium risk of bias in the replication or control criteria or three medium-risk-of-bias criteria,

- 'low' for remaining studies.

We considered a study to be unreliable, and therefore directly excluded it if there was a total absence of replication or control.

## Data extraction

Several experimental methodologies have been used in the literature in order to quantify CFF. Electrophysiological protocols are the most frequently used and often involve electroretinograms (ERG) or, alternatively electroencephalograms, which respectively measure the electrical response of the retina and brain to flickering light. Behavioural protocols have also been used and are mainly of two kinds: either a two-alternative forced choice procedure that requires an animal to choose the preferred light source between the flickering one (usually with increasing frequencies) and the continuous one [28]; or optomotor protocols where the nystagmus reflex of the head or eye of an animal is monitored through increasing flicker frequencies [46].

CFF data were extracted by two reviewers (ML and AL). AL checked all the extracted data by ML, which means that half of the corpus was double checked. Also, a test was conducted on a subsample of studies by two reviewers (RS and AL). As our goal was to determine the

maximum CFF at which animals could be impacted, if an article was to test different light intensities, we extracted the maximum CFF recorded by authors. If data were only presented on graphs, we extracted them using WebPlotDigitizer [47]. We also extracted metadata from each study: namely methods, either electrophysiological or behavioural, light sources, light intensities at which CFF were measured. We added a measure of the low, variable or high light exposure levels an animal is likely to experience in natural conditions (following the method from Inger *et al.* (2014) and Healy *et al.* (2013) [33, 34]). Briefly, a nocturnal or deep-sea foraging animal is likely to experience a low exposure to light compared to the high exposure of a diurnal or shallow water foraging animal. Cathemeral and crepuscular animals are considered to be exposed to variable quantities of light. Each species was associated with its taxonomic class [48], its trophic guild—i.e. herbivore, omnivore, carnivore—and a measure of its body size—i.e. very small, small, medium, large, very large. Categories were build thanks to data mainly provided by two available online trait databases [49, 50]. Animals with a body mass inferior to 1 g were arbitrarily considered to be very small, between 1 g and $10^3$ g small, between $10^3$ g and $10^4$ g medium, between $10^4$ g and $10^5$ g large, and superior to $10^5$ g very large. Each study was also assigned its critical appraisal risk of bias.

## Statistical analysis

To assess the influence of taxonomic classes, risks of bias, methods, light sources, light exposure levels, body sizes and trophic guilds on CFF, we implemented a linear mixed modelling approach.

Due to high collinearity between our variables, we split our analysis into two linear-mixed effect models (LMM). We first assessed whether CFF varied between taxonomic classes by using a LMM with species as the random term. Using a subset of means, medians, ranges of values, and extrapolated CFF values, we considered 7 sufficiently represented taxonomic classes, which corresponded to 151 CFF values. Model fits (residual vs. fit plots) were visually checked and CFF was square root-transformed to reach the best normality of residuals.

A second more comprehensive LMM was then fitted to assess the influence of risks of bias, methods, light sources, light exposure levels, body sizes and trophic guilds on CFF. This model also had a species random term as well as a taxonomic class random term to account for the non-independence of CFF across species and taxonomic classes. Due to the low availability of data on body masses and trophic guilds for Insecta and Malacostraca, both classes were discarded and the analysis carried out on a subset of 79 CFF values. Like for the first model, CFF was square root-transformed and model fits were visually checked. The most parsimonious model was selected based on the computation of the relative importance (RI) of each variable proposed by De Kort *et al.* (2021) [51].

To assess to what extent variations of CFF were explained by our second model, we computed $R_{GLMM(m)}^2$, the marginal $R^2$ representing the variance explained by the fixed effects and $R_{GLMM(c)}^2$, the conditional $R^2$ representing the variance explained by both fixed and random effects. All statistical analyses were carried out on R software (version 4.1.2). The LMM was computed with the package 'lmerTest' [52]. The most parsimonious model and $R^2$ were determined thanks to the package 'MuMIn' [53]. Graphs were customized thanks to the 'ggplot2' package [54].

## Results

### Screening process

The complete literature screening process is presented on Fig 1. All citations and their inclusion/exclusion on titles, abstracts and full-texts, associated with the reason in case of exclusion, are listed in S4 and S5 Files.

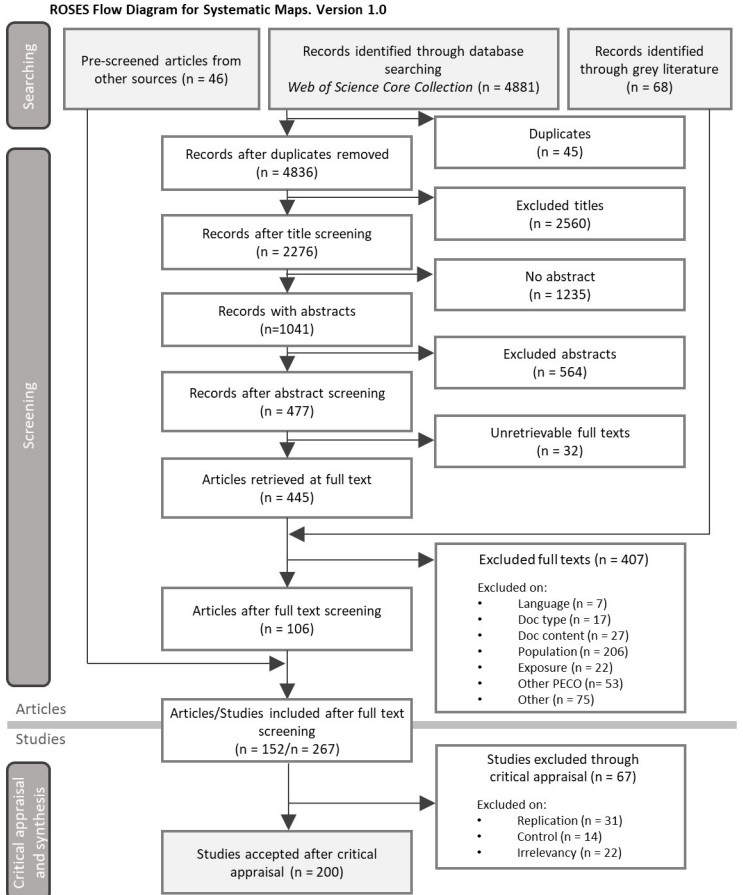

**Fig 1. ROSES flow diagram reporting the screening process of the articles and studies of the review** [55].

At first, 4881 citations were exported from the three databases and 2276 were selected on titles, among which 1041 had an available abstract. Abstracts screening lead to 477 accepted citations. Among them, 445 full-texts PDF were retrieved, of which 106 were kept after the full-text screening step. Over the whole screening process, 45 citations were identified as duplicates and were removed. Sixty-eight additional records came from our call for grey literature but none passed the full-text screening. Lastly, 46 other articles found by the review team that directly dealt with CFF [28, 33, 34, 42–45], but not directly found in the three considered databases, were added to the final corpus.

Overall, 152 articles were included and were split into 267 studies. After the exclusion of studies without any replication nor control, 200 studies were retained. The critical appraisal step resulted in 8% (16) of studies with a high risk of bias, 78% (155) with a medium risk of bias and 14% (29) with a low risk of bias (S6 File).

## Bibliometric results

Accepted articles were published between 1935 and 2020 with a major increase in the mid-1990s with less than 10 articles being published every five years before 1990 to more than 40 between 2015 and 2020 (S11 File). We classified studies' methodology into two groups: electrophysiology (153) and behavioural (47). In the first group, electroretinogram was the most frequently used methodology even if electroencephalograms and other physiological protocols

were also used. For the second group, two-alternative forced choice procedures were more often chosen, followed by optomotor ones. In both types of studies, authors used several types of light sources to produce their stimuli, LED being the most frequently chosen one (79) along with gas discharge lamps (37) and monochromators (33). More rarely, incandescent lamps and monitor screens were also used.

Fifteen taxonomic classes were found. Actinopterygii was the most represented taxon with more than 44 species being examined, followed by Malacostraca, Mammalia and Insecta with approximately 30 species (S11 File). Aves were studied in 23 species and Elasmobranchii as well as Reptilia were less examined both with 14 species. Only three Cephalaspidomorphi species were studied and, for each of the 7 remaining taxonomic classes, only one study was carried out. Overall, on the 200 CFF values that were observed, 156 different species were recorded. *Mus musculus* L. was the most examined species with 8 occurrences, along with *Gallus gallus* L. (6 studies) and *Rattus norvegicus* B. (6 studies). The vast majority of species (133) were studied only once.

## Distribution of CFF values

CFF was more often reported as a mean value (125 studies) but authors also used the range of values, the minimum or maximum, the median value or the extrapolation from their data. Among the 125 mean values, 77 were detailed by a measure of variation, either a standard error (65) or a standard deviation.

Reported values of CFF varied from a maximum of between 300 Hz and 500 Hz for the beetle *Melanophila acuminata* D. [56] to a mean (± SD) of 0.57 (± 0.08) Hz for the snail *Lissachatina fulica* B. [57].

Some commonly used light technologies such as LED or gas discharge (e.g. High Pressure Sodium) lamps may produce a flickering effect at a frequency of 100 Hz due to the 50 Hz electrical supply in Europe. For this reason, we considered a 100 Hz CFF threshold to identify species that might perceive ALAN's flicker in real *in-situ* conditions, outside at night. When only considering species living under low and variable light exposure levels, seven insects and two fish species among three taxonomic classes, namely Insecta, Aves, and Actinopterygii, had CFF higher than this threshold (Fig 2).

## Effects of taxonomy, experimental design characteristics and selected life history traits on CFF

The first LMM testing only for CFF variability between taxonomic classes, namely Insecta, Aves, Actinopterygii, Elasmobranchii, Malacostraca, Mammalia and Reptilia, was fitted on a subset of 123 species and 151 comparable CFF values. The effect of taxonomic classes was found to be highly significant (F-value = 28.11, df = 6, p-value < 0.001). Over the 7 taxonomic classes, Insecta had significantly higher CFF than Aves (p-value = 0.001), with respectively 150.6 (± 88.2) Hz and 89.4 (± 29.2) Hz. Both classes had also significantly greater CFF than the other five taxa (p-value < 0.007). Actinopterygii, Reptilia and Mammalia had similar CFF with respectively 51.5 (± 23.2) Hz, 45.0 (± 18.2) Hz and 41.6 (± 20.0) Hz. Even if not consistently significant, CFF from Malacostraca and Elasmobranchii were inferior with 27.2 (± 13.7) Hz and 26.6 (± 12.5) Hz (for all other comparisons between taxonomic classes see Fig 3 and Table 1).

The second model was fitted on a subset of 79 CFF values. The most parsimonious model only comprised the two fixed effects critical appraisal and light exposure levels (S11 File), which were both significant (F-value = 3.52, df = 2, p-value = 0.04; F-value = 11.74, df = 2, p-value < 0.001). Even though the majority of the variations of CFF was explained by random

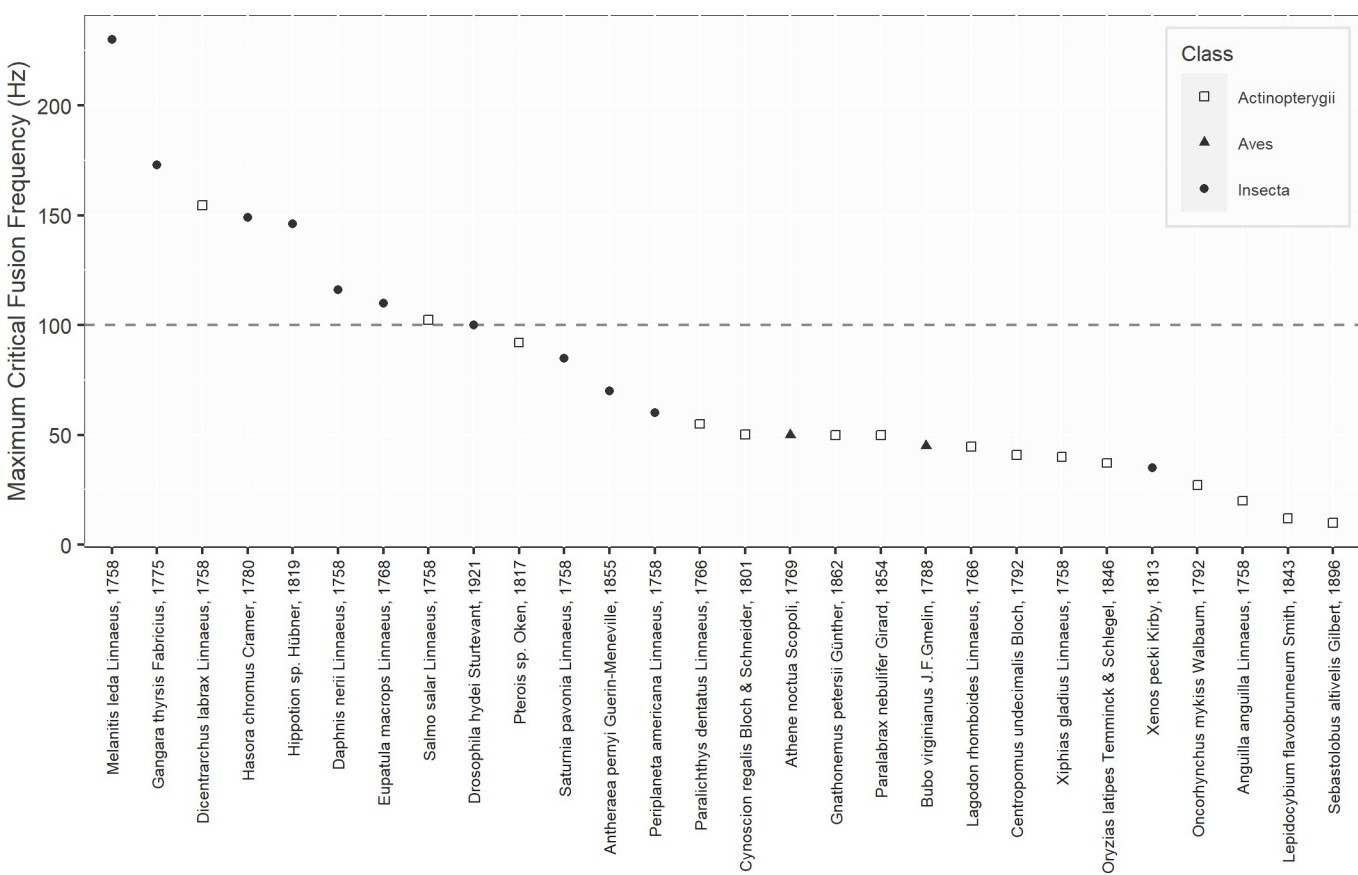

**Fig 2. Distribution of maximum Critical Fusion Frequencies (CFF).** Only species living under low and variable light exposure levels in Actinopterygii, Aves, and Insecta, the three classes that had species with CFF higher than 100 Hz, are represented. The dashed line represents the flicker frequency of a lamp on a 50 Hz electrical supply—i.e. 100 Hz.

terms, a significant proportion was still explained by the two fixed effects ($R_{GLMM(m)}^2$ = 0.28; $R_{GLMM(c)}^2$ = 0.72). CFF values provided by studies rated with a low risk of bias were significantly higher than those provided by medium risk of bias studies (p-value = 0.01) (Fig 4A). Animals exposed to variable and high levels of light had significantly higher CFF (p-value < 0.001) with a mean of 86.1 (± 30.8) Hz and 65.8 (± 29.5) Hz compared to 33.4 (± 16.9) Hz for animals exposed to low light levels (Fig 4B). Animals exposed to variable levels of light had also significantly higher CFF than those exposed to high light levels (p-value = 0.02).

## Discussion

Before the potential effects of flickering light on species can be evaluated, an essential prerequisite is to build a comprehensive knowledge base on CFF values—the latter being a first estimate of flicker perception by animals. We thus aimed at updating previous reviews on CFF [33, 34] by following a more comprehensive and transparent method of systematic reviews preconized by the CEE [37]. We identified that wide gaps of knowledge in CFF values remain to date for numerous taxa (e.g. Amphibia, Arachnida and some aquatic taxa) which should be filled by further experimental investigations.

When comparing our CFF values to those found by Inger *et al.* (2014) [33], the maximum CFF remained the same—i.e. between 300 Hz and 500 Hz for the beetle *Melanophila acuminata* D.—while a lower minimum was found—i.e. 0.57 (± 0.08) Hz for the snail *Lissachatina*

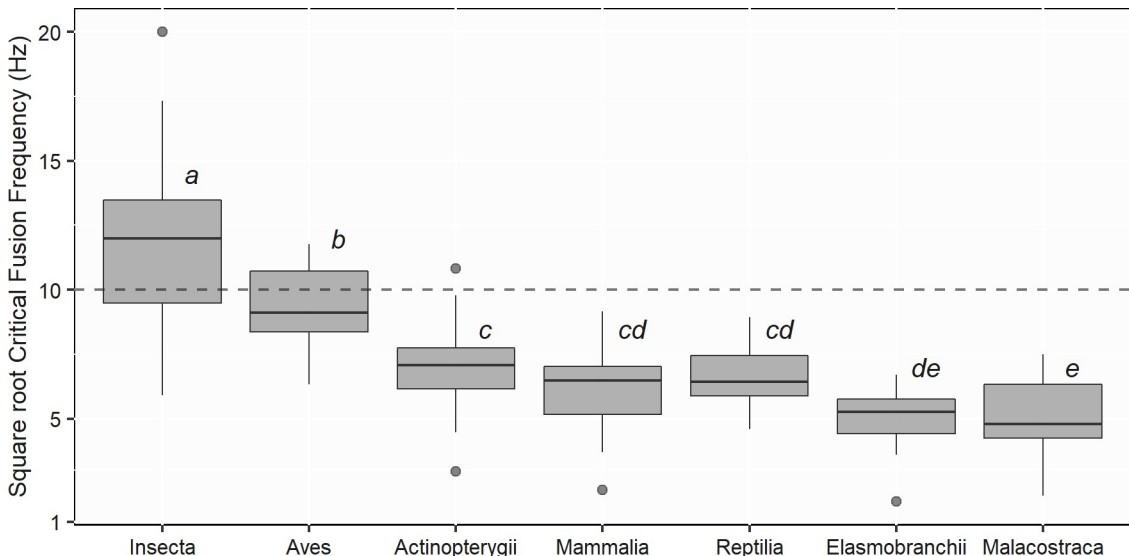

**Fig 3. Distribution of square root Critical Fusion Frequencies (CFF) across the most studied taxonomic classes.** The dashed line represents the flicker frequency of a lamp on a 50 Hz electrical supply—i.e. 100 Hz. Sample size: Insecta (n = 26), Aves (n = 17), Reptilia (n = 13), Actinopterygii (n = 35), Mammalia (n = 19), Malacostraca (n = 29), Elasmobranchii (n = 12). Linear mixed effect model significant differences (p-value < 0.05) are indicated by the letters above each boxplot.

*fulica* B.—which can be explained by the more extensive literature search strategy carried out here. Even though Hammer *et al.* (2001) [56] acknowledged that their measure of CFF from *M. acuminata*'s pit organ was not accurate, those two upper and lower boundaries for CFF have not been challenged for nearly 20 years. Thus, we can hypothesise that they represent solid limit frequencies within the animal kingdom.

When fitting similar models as Inger *et al.* (2014) [33] on our broader set of species, the similar overall results were observed in the present study: neither the methodology, electrophysiological or behavioural, nor the light source significantly explained differences in CFF, while taxonomic classes and light exposure levels did. Insecta had the greatest CFF followed by Aves, both of which had significantly higher CFF than the ones observed for all other classes. The ability to fly, which many birds and insects have, could explain their high critical

**Table 1. Linear-mixed models (LMM) results testing for differences of Critical Fusion Frequencies (CFF) between the most studied taxonomic classes.**

| | Insecta | | Aves | | Reptilia | | Actinopterygii | | Mammalia | | Elasmobranchii | | Malacostraca | |
|---|---|---|---|---|---|---|---|---|---|---|---|---|---|---|
| | p-value | | p-value | | p-value | | p-value | | p-value | | p-value | | p-value | |
| Insecta | - | | 1.44E-3 | ** | 4.87E-11 | *** | 1.31E-13 | *** | 9.82E-8 | *** | 7.37E-16 | *** | 4.49E-20 | *** |
| Aves | | | - | | 7.04E-4 | *** | 8.90E-4 | *** | 6.66E-3 | ** | 3.61E-7 | *** | 1.09E-8 | *** |
| Reptilia | | | | | - | | 0.46 | | 0.82 | | 0.05 | | 0.03 | * |
| Actinopterygii | | | | | | | - | | 0.73 | | 2.80E-3 | ** | 3.19E-4 | *** |
| Mammalia | | | | | | | | | - | | 0.06 | | 0.04 | * |
| Elasmobranchii | | | | | | | | | | | - | | 0.99 | |
| Malacostraca | | | | | | | | | | | | | - | |

Significant differences are indicated as follows:

*** p-value < 0.001,

** p-value < 0.01,

* p-value < 0.05.

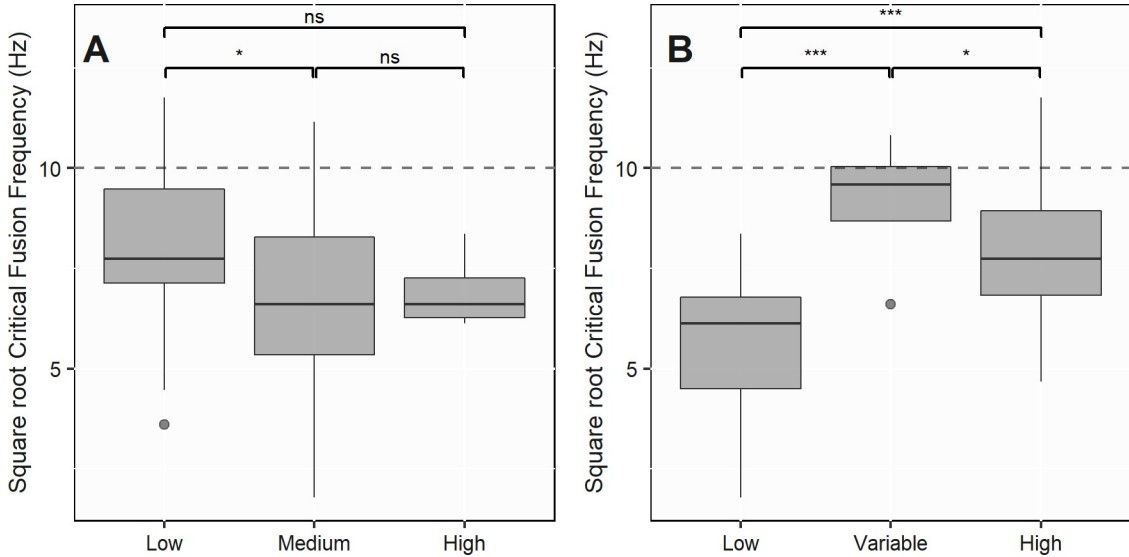

**Fig 4.** Square root Critical Fusion Frequencies (CFF) for (A) low, medium and high risks of bias and for (B) low, variable and high light exposure levels. The dashed line represents the flicker frequency of a lamp on a 50 Hz electrical supply—i.e. 100 Hz. Sample size: (A) Low (n = 20), Medium (n = 55), High (n = 4); (B) Low (n = 30), Variable (n = 4), High (n = 45). Linear mixed effect model differences are indicated as follows: *** p-value < 0.001, ** p-value < 0.01, * p-value < 0.05, ns non significant.

fusion frequencies since fast visual systems may be required to accomplish accurate manoeuvres and possibly avoid collisions [29, 33]. This latter finding is also interesting when considering the current decline of insect populations [58, 59] of which ALAN, alongside the loss of natural habitats and the use of pesticides [60], may be one of the main drivers [4, 5]. We can, therefore, hypothesise that flicker might represent a meaningful component of light pollution, along with intensity, spectral composition and temporality.

Even though we were not able to show any significant effect of trophic guilds on CFF with our model, the prey-predator relationship and its impact on CFF have already been discussed extensively in the literature. Indeed, Potier *et al.* (2019) [28] have linked the temporal resolution, defined as the ability to resolve rapid movements and estimated by the CFF, to the speed of the prey of diurnal raptors. They showed that the peregrine falcon, which hunts birds, had a higher temporal resolution than the Harris's hawk which eats mainly terrestrial mammals. Likewise, in clear water marine habitats, McComb *et al.* (2010) [30] related the higher temporal resolutions of bonnethead sharks to their shallow and bright reef habitats and their fast-moving prey compared to the lower temporal resolution of blacknose sharks, which may forage in deeper and dimmer water environments. Differences in temporal resolutions may even rely on a co-evolutionary relationship between the prey and the predator. Indeed, Boström *et al.* (2016) [29] argued that the high speed of insect prey could represent a major driver of high temporal resolutions in flycatchers, their bird predator, which could, then, drive those of insects further away.

Similar to our examination of trophic guilds, our model did not show any significant effect of body size on CFF. However, one could think that the smaller the animal, the higher the CFF might be (S11 File). Our analysis also showed that nocturnal animals had significantly lower CFF than crepuscular and diurnal ones, which is in line with the often reported trade-off between light sensitivity and temporal resolution—i.e. higher light levels are needed for high rates of sampling [30, 33, 34]. Those two previous results have already been assessed by Healy *et al.* (2013) [34] who demonstrated the correlation between body mass, metabolic rate and

temporal resolution—i.e. small animals with high metabolic rates in high light environments tend to have much higher CFF than large animals with low metabolic rate in darker environments.

One additional result provided by our modelling approach concerns the degree of confidence that should be granted to reported CFF values. Indeed, we found that CFF coming from low risk of bias studies were significantly greater than those from medium risk of bias studies and also possibly high risk of bias studies—the latter being not significant which may be due to the small sample size (only 4 studies). This result shows that, in order to correctly assess a species' CFF, a robust experimental protocol is needed and determines the level of confidence that can be granted to a CFF value. Then, replicating the experiment, repeating the measures, having a clear control, randomising the selection of individuals and appropriately preparing specimens before the experiment seem to directly influence the accuracy of CFF values. In addition, regarding the exposure criterion, studies experimenting on several increasing light intensities were more usually given low risk of bias that the ones using only one unique light intensity. Indeed, we considered that several increasing light intensities had to be trialled to ensure that a true maximum of FFF (i.e. the CFF) was reached.

## The perception of artificial anthropogenic light flicker

Due to the alternating nature of power supply, some anthropogenic light sources may flicker at a frequency of 100 Hz. This flicker has been thoroughly studied in fluorescent lighting [24, 33, 61, 62] but can be found in vapour discharge luminaires as well. While the latter are still used for outdoor lighting, it has hardly ever been the case for fluorescent lighting. Be that as it may, both types are increasingly being replaced by LEDs, that are proving much more energy-efficient. Various LED technologies have been engineered and each one may or may not show a 100-Hz flickering effect depending on the quality of their driver, the electronic component supplying power to the diode, and their dimming technology which relies, in the case of Pulse Width Modulation (PWM) technology, on rapid ON/OFF sequences at varying frequencies (usually 100 Hz–400 Hz) according to the chosen light intensity.

We thus considered the 100-Hz threshold to be relevant in order to assess the potential perception of anthropogenic flicker by animals. By isolating nocturnal and crepuscular animals, we were able to identify seven insects and two fish species which were much more likely to experience adverse effects from artificial light flicker. As we observed important intra-specific variation in CFF, we could also expect that some other species near the 100-Hz threshold were at risk of being impacted. Chatterjee *et al.* (2020) [63] provided the latest data on nocturnal and crepuscular insects CFF and looking at maximum values, they measured CFF of 230 Hz, 173 Hz and 149 Hz, respectively in the crepuscular butterflies *Melanitis leda* L., *Gangara thyrsis* F., *Hasora chromus* C., and 146 Hz, 116 Hz, 110 Hz in the moths *Hippotion* sp. H., *Daphnis nerii* L., and *Eupatula macrops* L., finally covering one of the poorly studied groups that Inger *et al.* (2014) [33] highlighted. Contrary to Inger *et al.* (2014) [33] who argued that the actual perception of flicker by animals should be limited—as they found that only diurnal animals were able to perceive flicker—we identified some crepuscular and nocturnal species that could, in fact, be influenced by flickering ALAN. Nonetheless, the actual perception of flicker—in real *in-situ* conditions—relies on more complex patterns and cannot be deduced solely by comparing species' CFF and light sources flicker frequencies. Indeed, the flicker perception depends also on light intensity and thus indirectly on the distance to the light source, as Inger *et al.* (2014) [33] stated. Flicker from an intense light source would indeed be perceptible from further away, hence the importance of keeping outdoor lighting at low light levels. In addition, the orientation of light sources could also be crucial because, as light emitted horizontally or

worse upwards can be spotted from farther away by animals, its flickering effect could be more perceptible as well.

Eventually, we would like to point out that animals may be subjected to the deleterious effects of flicker even though they cannot perceive it consciously. Indeed, Lu *et al.* (2012) [64] argued that a chromatic flicker at frequencies between 42.5 and 75 Hz, superior to the human CFF and therefore consciously unperceivable, was still able to entail their human subjects' alerting and orienting attentional networks. Even though a first assessment of CFF seems essential, such results could challenge their wider use and justifies the need for a potential future systematic review on the specific matter of the impacts of flashing and flickering light on biodiversity.

## Knowledge gaps

As was first noted by Inger *et al.* (2014) [33], this review highlights how an overall lack of data on animals' CFF remains and continues to hamper our understanding of the impacts of anthropogenic lighting on biodiversity. We, thence, reiterate the important need for additional primary research in order to improve our knowledge of CFF in as many species as possible. Indeed, on the millions of species that populate the Earth, just over 150 were studied thus far accounting for only 15 taxonomic classes. This scarcity of data is particularly noticeable in key nocturnal taxa, which are likely to be greatly disturbed by ALAN. As such, the flicker perception of many more nocturnal and crepuscular insect species should be studied as well as those of nocturnal birds and bats, two other groups on which no additional data was retrieved in our systematic review compared to previous reviews [33, 34].

Additionally, based on critical appraisal, we identified that very few studies achieved a low risk of bias according to the criteria defined in this review. For instance, a large number of authors failed to indicate if they had performed any repetition or randomisation. Likewise, very little information was provided on the state of animals prior to the experiments in some studies, which could be an important predictor of an animal's response to light treatments [65]. Too few studies in our corpus used several (or not as many as needed) light intensities in order to reach the true maximum flicker fusion frequency that is the CFF. Some studies could not report true maximum FFF as the highest frequency that their apparatuses could attain was too low for their specimens. In this case, we chose to report the highest FFF the authors could attain as a provisional CFF value. Such values should later be reassessed in order to build a more accurate CFF database. We also advocate the need for a strong and consistent reporting of CFF data. Indeed, accurate values of CFF should absolutely be reported as a mean attached to a measure of variation, which only one third of our studies provided, despite being a basic statistical need. Last but not least, we remind the essential need for replications and controls in order to measure reliable CFF values. Many recent studies have only used one specimen or have failed to report the use of a reference electrode for electroretinograms and had to be excluded after critical appraisal. While we acknowledge the difficulties in collecting and/or training large numbers of individuals, it is nonetheless critical to include measurements of multiple individuals for robust estimates of a species' threshold of flicker perception.

## Review limitations

Due to time limitation and financing constraints, we sometimes had to reduce our requirements comparing to CEE guidelines. First, we could not include search engines (e.g. Google Scholar) in our search strategy, nor could we request supplementary databases (e.g. Scopus) which would have probably increased the reliability of our search by increasing the number of test list articles indexed in the requested databases. In addition, after title screening, we had to

set aside citations that did not possess an appended abstract. Indeed, dealing with these citations would have meant searching for the full-texts of these 1235 citations, which represented an additional workload unfeasible within the scope of our project. However, this additional work remains feasible subsequently thanks to the S7 File which lists these citations without abstract. We also could not exploit the reviews (i.e. articles that might include some CFF values through their full-texts) identified through citation screening—except Inger *et al.* (2014) [33] and Healy *et al.* (2013) [34]—and we then collated them in S8 File for further exploitation. We are conscious that these shortcomings may lead to an underestimation of the number of identified CFF, which could potentially have been assessed in more species, and could provide less robust data as more values for a given species might have been found. Nevertheless, our database already provides many additional CFF values compared to previously published reviews [33, 34].

Lastly, studies were only critically appraised by one reviewer whereas CEE guidelines call for a double independent assessment by two reviewers. However, we performed a test between two reviewers on a subsample of articles before starting critical appraisal. Accordingly, we could not carry out a double independent data extraction on the whole corpus, as requested by the CEE, and only performed the double independent extraction on one half of the corpus.

## Conclusions

This systematic review, based on the method preconized by the CEE, provided a database of 200 critical fusion frequency values for 156 different species. This database complements the existing reviews on CFF by providing a wider range of more confident values. It, then, represents a current and qualitative state of knowledge of flicker perception in the animal kingdom, which could be helpful for scientists and researchers as well as for practitioners, such as lighting managers and designers. Here, we found that the beetle *Melanophila acuminata* D. could perceive the highest flicker frequency among all the recorded species whereas the snail *Lissachatina fulica* B. had the lowest temporal resolution. Insects and birds were the two taxa with the highest CFF. Moreover, nocturnal species had lower CFF than diurnal and crepuscular ones. We observed that some nocturnal and crepuscular insects could potentially perceive the flicker of anthropogenic light sources, which means that they could be subjected to its deleterious adverse effects. We also identified that ensuring the best experimental practices possible when determining CFF is crucial in order to report robust and accurate values of a species threshold of flicker perception. Nevertheless, the number of collected CFF values remains very low in relation to the number of all living animal species. In addition, this lack of data especially concerns key nocturnal taxa like birds and bats that are likely to be exposed to outdoor lighting. Therefore, the results presented here are necessarily provisional. We thus argue that many more species should be pressingly investigated by researchers to improve our knowledge of flicker perception by animals.

## Supporting information

**S1 File. ROSES reporting standards.**
(XLSX)

**S2 File. PRISMA checklist.**
(PDF)

**S3 File. Articles from the test list.**
(XLSX)

**S4 File. Citation screening.**
(XLSX)

**S5 File. Citations excluded at full-text.**
(XLSX)

**S6 File. Studies critical appraisal.**
(XLSX)

**S7 File. Citations without an abstract.**
(XLSX)

**S8 File. Reviews identified through screening.**
(XLSX)

**S9 File. Unobtainable articles at full-text.**
(XLSX)

**S10 File. CFF dataset.**
(XLSX)

**S11 File. Statistical analyses results.**
(DOCX)

## Acknowledgments

We would like to thank Simon Potier who enlightened us about behavioural and optomotor CFF discrimination protocols. We also wish to thank Françoise Viénot, Christophe Martin-sons, Jack Falcón, Christian Kerbiriou, Léa Mariton and Matthieu Iodice for providing us with literature, as well as Dakis-Yaoba Ouédraogo and Simon Blanchet for their fruitful statistical advice which contributed to substantially improve the quality of our manuscript. At last, we sincerely thank Marie-Pierre Alexandre for her support and advice all along our study.

## Author Contributions

**Conceptualization:** Romain Sordello, Virginie Nicolas, Gaël Obein, Yorick Reyjol.

**Data curation:** Alix Lafitte, Marc Legrand.

**Formal analysis:** Alix Lafitte.

**Investigation:** Alix Lafitte, Romain Sordello, Marc Legrand.

**Project administration:** Romain Sordello, Yorick Reyjol.

**Writing – original draft:** Alix Lafitte.

**Writing – review & editing:** Romain Sordello, Marc Legrand, Virginie Nicolas, Gaël Obein, Yorick Reyjol.

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
