## [Decision Letter · Decision Letter 0]

30 Aug 2022

PONE-D-22-13677A Flashing Light may not be that Flashy: a Systematic review on Critical Fusion FrequenciesPLOS ONE

Dear Dr. Lafitte,

Thank you for submitting your manuscript to PLOS ONE. After careful consideration, we feel that it has merit but does not fully meet PLOS ONE’s publication criteria as it currently stands. Therefore, we invite you to submit a revised version of the manuscript that addresses the points raised during the review process.

Two reviewers have examined this manuscript and both contributed important comments. Reviewer 1 has some strong comments about the organization of the manuscript and suggests that the authors could focus more on answering ecologically or taxonomically motivated questions in addition to compiling the data on CFF. I tend to agree somewhat with Reviewer 1's comments, but also note that this is not a requirement for publication. Consequently, I will leave it to the authors to decide how much revision they would like to do in this respect. Reviewer 2 provides many good comments that should be considered.

I also include a file with tracked changes in which I identified a small number of typographical or grammatical errors.

We look forward to receiving your revised manuscript.

Kind regards,

Christopher Nice, Ph.D.

Academic Editor

PLOS ONE

Journal Requirements:

Reviewers' comments:

Reviewer's Responses to Questions

**Comments to the Author**

1. Is the manuscript technically sound, and do the data support the conclusions?

Reviewer #1: Partly

Reviewer #2: Yes

2. Has the statistical analysis been performed appropriately and rigorously? 

Reviewer #1: No

Reviewer #2: Yes

3. Have the authors made all data underlying the findings in their manuscript fully available?

Reviewer #1: Yes

Reviewer #2: Yes

4. Is the manuscript presented in an intelligible fashion and written in standard English?

Reviewer #1: Yes

Reviewer #2: No

5. Review Comments to the Author

Reviewer #1: This paper is focussed on the observed phenomenon that animals vary in their ability to detect flashing lights and this trait has potential behavioural and hence possible conservation relevance. This paper is effectively a description of a dataset; albeit one that has been collected very rigorously. While there is some analyses of the data presented, there is little by way of a hypothesis driven approach and ultimately the analyses performed are not well directed or executed. There have been previous studies on what might explain variation in this trait among animals, but this current paper does not clearly develop those findings and does not provide obvious new discoveries.

The paper swaps a lot between stating that its main aim is to publish a rigorous dataset or that its aim is to describe the potential impacts of this trait on biodiversity and light pollution. Ultimately i think this is really a dataset description at its core and the value to understanding variation in the trait is secondary. My advice would be to either focus on the dataset itself and submit for publication in a journal such as Nature Publishing Group's journal Scientific Data https://www.nature.com/sdata/publish or Elsevier's Data in Brief https://www.journals.elsevier.com/data-in-brief . The alternative is to go back and formulate some clear questions that can be answered using this dataset, that are based on sound logical argument based on the literature and make these objectives the key focus of the narrative.

My justification for this assessment is based on various statements in the abstract and introduction.

**Abstract**

The stated objective is "This review aims at collecting CFF values for as many animal species as possible through a comprehensive, transparent and replicable systematic literature survey according to the Collaboration for Environmental Evidence standards." This says that the paper is about a dataset, and makes no mention of testable hypotheses.

The stated methods are entirely around collation of the dataset.

The results start to bring in some patterns of variation among taxonomic groups or individuals species, but this comes out of nowhere based on the previous content in the abstract.

**Introduction**

It is stated that artificial light at night (ALAN) affects species and ecosystems, but no mechanism is explored (ultimate effects are discussed though), and hence does not set the scene for hypotheses about what factors might explain variation in CFF as a trait.

Statements are made that new technologies such as LEDs can have an impact, but again no mechanisms or details are provided. Do LEDs flicker? are they more intense? etc...

The new study is stated as an improvement over Inger et al 2014 and Healy et al 2013. Certainly the data collection aspect of the current study is more comprehensive and rigorous than these two papers, but they are not "reviews" as stated in the text as they both aimed to test actual hypotheses. Their aim was not to collate a dataset but rather draw some inference on the drivers of CFF in specific groups. This comparison and criticism is perhaps not entirely warranted and should be rephrased.

The introduction concludes with a statement "This work focused on wild and domestic animals, excluding humans, and aimed at answering the following review question: until which frequency a species can perceive flicker?" which is not a well stated question or hypothesis. It ultimately comes down to stating that something was measured and the aim is to right down those numbers. Maybe the aim is to identify which species are perhaps most likely to be affected by ALAN but even if that is the case it is not clear how exactly that assessment can be made using the data in question because sensitivity to flickering light at 50Hz is not the same as negatively impacted by same.

**Material & Methods**

The approach is very rigorous and well described. While the authors point to potential biases in not following a rigorus protocol such as the one they have adopted, they do rather casually state that only articles written in English and French were considered. This of course could be a source of bias no different to the others they identify and should be acknowledged as such.

The critical appraisal section is very rigorous and welcome.

Healy et al 2013 showed a clear trend of log(CFF) with log(body mass), but here the authors choose to collapse body size into large, average and small. There is no detailed description in the main text for how these categories were defined and decided, and it seems like a big missed opportunity to just include body mass, which is known to be a key trait in nearly all studies of within and between species variation.

It is not clear from the main text what variables were included as random factors and which were fixed factors in the models. One aspect of Healy et al 2013 that was a key feature of analysing variation i CFF between species as the inclusion of a phylogenetic random term but this appears to not be included in the present study. Some discusison of this is at least warranted. Presumably the main issue might be a lack of a good tree for insects?... but it seems like a missed opportunity to simply ignore this aspect.

Square root transforming the CFF data seems a bit arbitrary. Is there a mechanistic reason to do so? There might be a mechanistic reason to go for 1/CFF which would be 1/Hz which is wavelength and hence a measure of the integration window of perception. Equally Healy et al 2013 argued for modelling log(CFF) ~ log(body mass) as a logically argued allometric scaling relationship.

**Results**

Why was one of the criteria that a paper had to have an abstract?

The recording of whether a study was electrophysiological or behavioural is welcome.

The threshold of 100Hz is not well explained. That is it is not clearly explained how potential perception of ALAN's flicker is linked to CFF or indeed how one might prove an impact.

Running a LMM, not getting the result you expected and then running a random forest is not a rigorous way to perform an analysis and risks fishing for results. What about all the other approaches we could have used? It seems a shame to have gone to good detail on compiling a rigorous dataset only to throw multiple statistical approaches including model selection and random forests at the data without a well argued reason.

**Conclusions**

Ultimately the conclusions support my sense that this paper is primarily about compilation of a rigorous dataset and does not test well argued questions or hypotheses about the variation in this trait. As such i feel that it would be more naturally published as a dataset and not as a research article.

Reviewer #2: In this interesting paper the authors have compiled a comprehensive database of CFF in the animal kingdom. Importantly, they have used a rigorous process following standardised guidelines to compel the database and they have detailed the entire process. Then, using this database, which I believe is the largest and most comprehensive of its type, they performed analyses to (1) identify ecological correlates of CFF (such as body size, activity patterns and environmental light levels, and trophic guild and (2) to identify species that may be particularly at risk from flicker caused by anthropogenic lighting, assuming a critical frequency of 100 Hz. The latter is particularly important because anthropogenic light pollution is increasing and there is strong evidence that it is having detrimental effects on ecosystems worldwide. Using their large database, the authors have also been able to identify knowledge gaps such as amphibians and nocturnal aerial species like bats, birds, and insects. I enjoyed reading this manuscript. However, my main concerns are to do with how the authors assigned different ecological categories to each of the species in their database. At the very least I think they need to provide more information onto how the categories were defined and justify trying to fit should a brough range of animals representing the entire animal kingdom onto such simple 3- or 4-point scales of, for example, body size and tropic level/guild. I also found numerous examples of grammatical errors that need to be addressed, and some passages of text where I do simply not understand what the author s mean by what they have written.

Abstract

• Change “Insects and birds had higher CFF than all other taxa studied whereas nocturnal species had lower CFF than diurnal and crepuscular ones.” to “Insects and birds had higher CFFs than all other taxa studied. Irrespective of taxon, nocturnal species had lower CFF than diurnal and crepuscular ones.”

• Change “We also found that primary consumer might have greater CFF than species from higher levels of the food chain.” to “We also found that primary consumers might have greater CFFs than species from higher levels of the food chain.”

Introduction

Page 4

• Change “anthropogenic driver behind insect decline” to “anthropogenic drivers behind insect decline”.

• Change “and have been linked” to “and has been linked”.

• Change “At last, ALAN could also” to “In addition, ALAN could also”.

• Change “disrupting two key ecosystem services that are pollination and seed dispersal” to “disrupting two key ecosystem processes, namely pollination and seed dispersal”.

• Change “Light impacts on biological organisms have been linked to several key components of lighting,” to “Impacts on biological organisms have been linked to several key components of artificial lighting,”.

Page 5

• Change “as it lowered the number of captured Diptera, Hemiptera and Lepidoptera individuals.” to “as lowered numbers of Diptera, Hemiptera and Lepidoptera were caught in traps associated with a flickering light.”

• Change “Species very perception” to “Species perception”

• Change “build a better knowledge on CFF distribution in the animal kingdom. In this purpose,” to “gain better knowledge of variation in CFF across the animal kingdom. To this end, “.

Page 6

• Change “may then be” to “may thus be”.

• Change “we propose a systematic review on” to “we present a systematic review of”.

• I do not understand what the research question “until which frequency a species can perceive flicker?” means. Can this be rephrased please?

Materials and Methods

Page 10

• Change “which both measure the electrical response of the retina or brain to flickering light.” to “which measure the electrical response of the retina or brain to flickering light, respectively.”

• Change “require an animal” to requires an animal”.

Page 11

• I have concerns about how species were categorised based on trophic status. In the CFF database (file S10) species are coded on a four-point scale from 0-3. However, in the text the authors state that species were classified as being a primary consumer, omnivorous (this should be omnivore and predator). How do these three categories match onto the four-point scale of 0-3? Also, what constitutes a predator? Does this category include secondary and tertiary consumers? How were different species assigned into these ecological categories? Was this based on the literature, or intuition, or the authors own knowledge? Did the authors try to further separate predators into secondary and tertiary consumers? Having tried to classify large datasets of species into different trophic level I do appreciate that it is very difficult to classify such a wide range of animals, but I wonder if using such a simplistic scale means that important biological information is lost here. For example, all of the elasmobranchs within the database are classified as having the same trophic level (3). However, although they are all predators, they actually operate on different trophic levels in aquatic food webs. For example, rays and guitarfish feed on invertebrates and can be considered secondary consumers, while sharks such as scalloped hammerhead sharks, feed on rays and so are tertiary consumers. In addition, although not included in this study, there are other apex predator species of shark that occupy the roles of quaternary consumers, that will feed on both rays and scalloped hammerhead sharks.

• I also think the authors need to provide more information and clarity about how all of the species in the data based were classified in terms of body size. In a study dealing with so many different forms of animal life ranging across several orders of magnitude, how is it possible to divide them into large, average and small? For a species to be grouped into one of these three categories, did it have to be above or below a certain body size, for example? Also, what does the average category mean? Average body size in respect to what? All species in the animal kingdom, species included in this study? Perhaps the authors mean small, mid-size and large as opposed to small, average and large? More information is required. I also find it strange that whatever scheme the authors have used has resulted in, for example, all of the insects and birds being grouped in the same category (small), even though a house fly may be a few mm in size, weighing a fraction of a gram, whereas a Harris hawk can weigh 800-1000g and have a wingspan of over 1 metre. The system used also means that the Great horned owl, which is a very large bird with a wingspan of over 1 metre is classified as ‘small’, whereas a domestic cat, which weighs more than a great horned owl, but which is smaller in terms of body size/length, as classified as ‘average’. Another example is that a trout (30-60 cm, 05-3 kg), cuttlefish (30-40 cm, 2-4 kg) and sheep (50-100 kg) are all considered to be ‘average’. At the very least the authors need to better define and quantify their body size categories, and I think they should potentially consider have a broader range of categories.

Results

Page 13

• Change “two-force choice procedures” to “two-alternative forced choice procedure”.

• I do not understand the rationale for presenting the CFF values a for a species in the text, both on page 13 and throughout the manuscript. Why is the mean value plus/minus standard deviation or some other measure of variability enclosed within parentheses? For example, I would change “to (0.57 ± 0.08) Hz for the snail Lissachatina fulica B.” to “to 0.57 (± 0.08) Hz for the snail Lissachatina fulica B

• Results in general. I think the results and analysis are fine as the manuscript stands, but if the authors end up revising their categorises for body size and trophic level, for example, then this will have an impact on the results and statistical analysis.

Discussion

Page 16.

• The results show that insects and birds have the highest CFF values. A logical next step for me at least would be to think that there is some correlation between needing a faster visual system and the sensory demands of flight. However, the authors do not seem to even mention this. Are flying animals more or less likely to be exposed to anthropogenic flickering light sources?

Page 17

• In the discussion of CFF in sharks the authors mention that McComb et al. hypothesized that variation in CFF between bonnethead, scalloped hammerhead and blacknose sharks could be because bonnethead and scalloped hammerhead sharks (which have relatively higher CFFs) live in shallow and bright reef habitats and feed on their fast-moving prey compared blacknose sharks, which usually forages in deeper and dimmer water environments. However, in table S10 the authors have classified bonnetheads as experiencing ‘high’ light exposure while both scalloped hammerheads and blacknose sharks are classified as having ‘low’ light exposure. It seems that the light level classifications assigned to these species contradicts the primary source of information. It also seems strange that the authors have highlighted these sharks as an example when their own coding of these specie sin their database appears to contradict the primary reference source.

• Change “Our data showed that primary consumer” to “Our data showed that primary consumers”.

• Change “Such assumptions should, nevertheless, be studied” to “Such assumptions should be studied”.

Page 18

• Change “When the latter are still used for outdoor lighting” to “While the latter are still used for outdoor lighting”.

• What does “Chatterjee et al., (2020) [61] brought the latest data” mean? What do the authors mean by ‘brought’ in this context?

Page 19

• Change “For as crucial assessing an animal actual perception of flicker is” to “As crucial as assessing an animals actual flicker perception is”.

• Change “As it was first noted by Inger et al. (2014) [33] and even if more and more studies have been carried out lately, an important lack of data on animals’ CFF remains and is hampering our understanding of the impacts of anthropogenic lighting on biodiversity” to “As was first noted by Inger et al. (2014) [33], this study highlights how an overall lack of data on animals’ CFF remains and continues to hamper our understanding of the impacts of anthropogenic lighting on biodiversity.”

Page 20

• Change “There, nevertheless, remains many recent studies using only one specimen or failing to report the use of a reference electrode for electroretinograms” to “Many recent studies have only used one specimen or have failed to report the use of a reference electrode for electroretinograms”.

Page 22

• Change “outdoor lightings. Then, this result is necessarily provisional.” to “outdoor lighting. Therefore, the results presented here are necessarily provisional.”

6. PLOS authors have the option to publish the peer review history of their article (what does this mean?). If published, this will include your full peer review and any attached files.

Reviewer #1: No

Reviewer #2: No

---

## [Author Response · Author response to Decision Letter 0]

28 Oct 2022

Editor: 

Two reviewers have examined this manuscript and both contributed important comments. Reviewer 1 has some strong comments about the organization of the manuscript and suggests that the authors could focus more on answering ecologically or taxonomically motivated questions in addition to compiling the data on CFF. I tend to agree somewhat with Reviewer 1's comments, but also note that this is not a requirement for publication. Consequently, I will leave it to the authors to decide how much revision they would like to do in this respect. Reviewer 2 provides many good comments that should be considered.

I also include a file with tracked changes in which I identified a small number of typographical or grammatical errors.

=> We thank you and the reviewers for your interest in our manuscript. We detail here how we have integrated the requests. All revisions to the manuscript by the editor were taken into account in this new version.

=> Based on comments from Reviewer #2, we rewrote some parts of our manuscript to better answer the ecological questions at the heart of this work. Notably, we rephrased our ecological questions: “what evidence exists regarding animal critical fusion frequencies? What is the distribution of CFF between species? Which species are more at risk of being impacted by artificial lighting flicker?” (p. 6-7)

=> Overall, we decided to improve the quality of our database and statistical analyses. First, we built a more accurate database for trophic guild (renamed to better fit our new categories) and body mass by extracting data on as many species as possible on online available trait databases - mainly FishBase (www.fishbase.org) and Animal Diversity Web (https://animaldiversity.org). We also more clearly defined how we classified species (into 5 categories instead of 3) as being Very small, Small, Medium, Large or Very Large weight categories. In addition, we chose to only consider 3 levels of trophic guilds that are Herbivore, Omnivore and Carnivore.

=> Due to the lower availability of data for trophic guilds and body masses notably for Malacostraca and Insecta and also based on the comments from Reviewer #1, we improved our modelling approach. First, we dropped our RandomForest to only focus on the results from our LMMs. To take into account collinearity, we decided to split our linear-mixed modelling approach in two by first assessing the effect of taxonomic classes alone. We also added taxonomic class as a random factor in the second model to account for phylogenetic non-independence between CFF values. We added this new methodology to the ‘Material & Methods’ section and corrected the ‘Results’ and ‘Discussion’ sections accordingly, notably concerning critical appraisal and taxonomic classes significant comparisons. (p. 11-12 and 16-19 and 21)

=> We corrected Fig 1 and the main text after spotting a misclassified duplicate that altered the reported the reported volumes of citations. We also changed our additional files to improve conciseness (see S11 File).

Reviewer #1: 

This paper is focussed on the observed phenomenon that animals vary in their ability to detect flashing lights and this trait has potential behavioural and hence possible conservation relevance. This paper is effectively a description of a dataset; albeit one that has been collected very rigorously. While there is some analyses of the data presented, there is little by way of a hypothesis driven approach and ultimately the analyses performed are not well directed or executed. There have been previous studies on what might explain variation in this trait among animals, but this current paper does not clearly develop those findings and does not provide obvious new discoveries.

The paper swaps a lot between stating that its main aim is to publish a rigorous dataset or that its aim is to describe the potential impacts of this trait on biodiversity and light pollution. Ultimately i think this is really a dataset description at its core and the value to understanding variation in the trait is secondary. My advice would be to either focus on the dataset itself and submit for publication in a journal such as Nature Publishing Group's journal Scientific Data https://www.nature.com/sdata/publish or Elsevier's Data in Brief https://www.journals.elsevier.com/data-in-brief . The alternative is to go back and formulate some clear questions that can be answered using this dataset, that are based on sound logical argument based on the literature and make these objectives the key focus of the narrative.

=> Based on these remarks, we tried to better phrase our questions. Our questions are now phrased: “what evidence exists regarding animal critical fusion frequencies? What is the distribution of CFF between species? Which species are more at risk of being impacted by artificial lighting flicker?” (p. 6-7)

My justification for this assessment is based on various statements in the abstract and introduction.

**Abstract**

The stated objective is "This review aims at collecting CFF values for as many animal species as possible through a comprehensive, transparent and replicable systematic literature survey according to the Collaboration for Environmental Evidence standards." This says that the paper is about a dataset, and makes no mention of testable hypotheses.

=> Agreed, we added the lacking ecological hypotheses: “what evidence exists regarding animal CFF? What is the distribution of CFF between species? Which species are more at risk of being impacted by artificial lighting flicker?” (p. 2)

The stated methods are entirely around collation of the dataset.

=> Agreed, we added a sentence on the statistical analyses we carried out “All relevant data were extracted and analysed to determine the distribution of CFF in the animal kingdom and the influence of experimental designs and species traits on CFF” (p. 2)

The results start to bring in some patterns of variation among taxonomic groups or individuals species, but this comes out of nowhere based on the previous content in the abstract.

=> We hope that previous additions answer that comment.

**Introduction**

It is stated that artificial light at night (ALAN) affects species and ecosystems, but no mechanism is explored (ultimate effects are discussed though), and hence does not set the scene for hypotheses about what factors might explain variation in CFF as a trait.

=> To improve the conciseness, clarity and understandability of the introduction, we had decided to address this topic and discuss it based on our results in the section ‘Discussion’. Indeed, we did not want to have an extended and unclear introduction.

Statements are made that new technologies such as LEDs can have an impact, but again no mechanisms or details are provided. Do LEDs flicker? are they more intense? etc...

=> Same as above, we decided to talk about ALAN flicker in the ‘Discussion’ section. First drafts of our introduction were considered too long and therefore its understandability was minimal.

The new study is stated as an improvement over Inger et al 2014 and Healy et al 2013. Certainly the data collection aspect of the current study is more comprehensive and rigorous than these two papers, but they are not "reviews" as stated in the text as they both aimed to test actual hypotheses. Their aim was not to collate a dataset but rather draw some inference on the drivers of CFF in specific groups. This comparison and criticism is perhaps not entirely warranted and should be rephrased.

=> As both articles collected data from numerous publications (and even applied a literature search strategy similar to ours in the case of Inger et al. (2014)), we considered them as reviews. We rephrased in the MS: “For decision-makers to be informed in the best way possible and to reach highly beneficial biodiversity protective actions, we argue that a more comprehensive and transparent literature survey is therefore needed”. (p. 6)

The introduction concludes with a statement "This work focused on wild and domestic animals, excluding humans, and aimed at answering the following review question: until which frequency a species can perceive flicker?" which is not a well stated question or hypothesis. It ultimately comes down to stating that something was measured and the aim is to right down those numbers. Maybe the aim is to identify which species are perhaps most likely to be affected by ALAN but even if that is the case it is not clear how exactly that assessment can be made using the data in question because sensitivity to flickering light at 50Hz is not the same as negatively impacted by same.

=> After comments in the abstract, we decided to improve the way we phrased our review questions: “This work focused on wild and domestic animals, excluding humans, and aimed at building a comprehensive knowledge base on animals’ CFF as well as answering the following questions: what evidence exists regarding animal critical fusion frequencies? What is the distribution of CFF between species? Which species are more at risk of being impacted by artificial lighting flicker?” (p. 6-7)

**Material & Methods**

The approach is very rigorous and well described. While the authors point to potential biases in not following a rigorus protocol such as the one they have adopted, they do rather casually state that only articles written in English and French were considered. This of course could be a source of bias no different to the others they identify and should be acknowledged as such.

=> Agreed, such bias exists in this work but it is acknowledged in the Collaboration for Environmental Evidence standards. We added a sentence on the matter: “We acknowledge that only including articles in those two languages constitutes a potential bias to our systematic review but this could not be avoided based the linguistic competences of the review team” (p. 8)

The critical appraisal section is very rigorous and welcome.

=> We are deeply thankful for this comment.

Healy et al 2013 showed a clear trend of log(CFF) with log(body mass), but here the authors choose to collapse body size into large, average and small. There is no detailed description in the main text for how these categories were defined and decided, and it seems like a big missed opportunity to just include body mass, which is known to be a key trait in nearly all studies of within and between species variation.

=> Based on general comments from Reviewer #2, we decided to improve the quality of our database by extracting body mass values from two online trait databases. However, due to heterogeneity on data types (either means, ranges of values or maximums) between the different databases, we decided to collapse body mass into 5 body sizes categories. This allowed to have the largest dataset possible for our modelling approach. As very little data could be found for Insecta and Malacostraca, they had to be discarded from our analyses.

=> We added explanations in the section ‘Material & Methods’ on how our categories were built and how it changed our statistical analyses. (p. 11-12)

It is not clear from the main text what variables were included as random factors and which were fixed factors in the models. One aspect of Healy et al 2013 that was a key feature of analysing variation i CFF between species as the inclusion of a phylogenetic random term but this appears to not be included in the present study. Some discusison of this is at least warranted. Presumably the main issue might be a lack of a good tree for insects?... but it seems like a missed opportunity to simply ignore this aspect.

=> Based on this comment, we decided to implement a taxonomic class random term in our modelling approach (See answer to Editor). We added: “This model also comprised a species random term as well as a taxonomic class random term to account for the non-independence of CFF values between taxonomic classes.” (p. 12)

Square root transforming the CFF data seems a bit arbitrary. Is there a mechanistic reason to do so? There might be a mechanistic reason to go for 1/CFF which would be 1/Hz which is wavelength and hence a measure of the integration window of perception. Equally Healy et al 2013 argued for modelling log(CFF) ~ log(body mass) as a logically argued allometric scaling relationship.

=> We used the square root transformation of CFF data as, in our case, it achieved the best normality of residuals compared to the log transformation preferably chosen by Healy et al. (2013) or Inger et al. (2014). This is classically carried out in modelling approach.

**Results**

Why was one of the criteria that a paper had to have an abstract?

=> We were confronted to a large number of citations without an appended abstract. Due to time constraints, we could not retrieve all full-texts associated to these citations and screen them. To do this would have represented an additional workload of 1235 full-texts to search and screen which was not possible within the scope of our funded project. We therefore had to put them aside. We however stress that such citations are clearly identified thanks to Supplementary file 7 to allow any person to undertake this work in order to complete the review.

The recording of whether a study was electrophysiological or behavioural is welcome.

=> We are deeply thankful for this comment.

The threshold of 100Hz is not well explained. That is it is not clearly explained how potential perception of ALAN's flicker is linked to CFF or indeed how one might prove an impact.

=> We tried to rephrase that sentence to improve its understandability: “As some commonly used light technologies such as LED or gas discharge lamps (e.g. High Pressure Sodium) may produce a flickering effect at a frequency of 100 Hz due to the 50 Hz electrical supply in Europe. For this reason, we considered a 100 Hz CFF threshold to identify species that might perceive ALAN’s flicker in real conditions outside at night.”. (p. 16)

=> Such results are later discussed in the ‘Discussion’ section.

Running a LMM, not getting the result you expected and then running a random forest is not a rigorous way to perform an analysis and risks fishing for results. What about all the other approaches we could have used? It seems a shame to have gone to good detail on compiling a rigorous dataset only to throw multiple statistical approaches including model selection and random forests at the data without a well argued reason.

=> Yes, we agree. Based on this comment, we decided to only keep our LMM and not to take into account the RandomForest anymore. We reformulated both the ‘Results’ and ‘Discussion’ sections accordingly. (p. 17 and 20)

**Conclusions**

Ultimately the conclusions support my sense that this paper is primarily about compilation of a rigorous dataset and does not test well argued questions or hypotheses about the variation in this trait. As such i feel that it would be more naturally published as a dataset and not as a research article.

=> As we believe this paper also aims at identifying CFF variations between species and which species are more at risk of being impacted by flicker, we rephrased our hypotheses in the ‘Abstract’ and ‘Introduction’ sections. (p. 6-7)

Reviewer #2: 

In this interesting paper the authors have compiled a comprehensive database of CFF in the animal kingdom. Importantly, they have used a rigorous process following standardised guidelines to compel the database and they have detailed the entire process. Then, using this database, which I believe is the largest and most comprehensive of its type, they performed analyses to (1) identify ecological correlates of CFF (such as body size, activity patterns and environmental light levels, and trophic guild and (2) to identify species that may be particularly at risk from flicker caused by anthropogenic lighting, assuming a critical frequency of 100 Hz. The latter is particularly important because anthropogenic light pollution is increasing and there is strong evidence that it is having detrimental effects on ecosystems worldwide. Using their large database, the authors have also been able to identify knowledge gaps such as amphibians and nocturnal aerial species like bats, birds, and insects. I enjoyed reading this manuscript. However, my main concerns are to do with how the authors assigned different ecological categories to each of the species in their database. At the very least I think they need to provide more information onto how the categories were defined and justify trying to fit should a brough range of animals representing the entire animal kingdom onto such simple 3- or 4-point scales of, for example, body size and tropic level/guild. I also found numerous examples of grammatical errors that need to be addressed, and some passages of text where I do simply not understand what the author s mean by what they have written.

Abstract

• Change “Insects and birds had higher CFF than all other taxa studied whereas nocturnal species had lower CFF than diurnal and crepuscular ones.” to “Insects and birds had higher CFFs than all other taxa studied. Irrespective of taxon, nocturnal species had lower CFF than diurnal and crepuscular ones.”

=> Modified. (p. 2)

• Change “We also found that primary consumer might have greater CFF than species from higher levels of the food chain.” to “We also found that primary consumers might have greater CFFs than species from higher levels of the food chain.”

=> Based on statistical analyses modifications, we decided not to report this result in this new version. (p. 2)

Introduction

Page 4

• Change “anthropogenic driver behind insect decline” to “anthropogenic drivers behind insect decline”.

=> Modified. (p. 4)

• Change “and have been linked” to “and has been linked”.

=> Modified. (p. 4)

• Change “At last, ALAN could also” to “In addition, ALAN could also”.

=> Modified. (p. 4)

• Change “disrupting two key ecosystem services that are pollination and seed dispersal” to “disrupting two key ecosystem processes, namely pollination and seed dispersal”.

=> Modified. (p. 4)

• Change “Light impacts on biological organisms have been linked to several key components of lighting,” to “Impacts on biological organisms have been linked to several key components of artificial lighting,”.

=> Modified. (p. 4)

Page 5

• Change “as it lowered the number of captured Diptera, Hemiptera and Lepidoptera individuals.” to “as lowered numbers of Diptera, Hemiptera and Lepidoptera were caught in traps associated with a flickering light.”

=> Modified. (p. 5)

• Change “Species very perception” to “Species perception”

=> Modified. (p. 5)

• Change “build a better knowledge on CFF distribution in the animal kingdom. In this purpose,” to “gain better knowledge of variation in CFF across the animal kingdom. To this end, “.

=> Modified. (p. 5)

Page 6

• Change “may then be” to “may thus be”.

=> Modified. (p. 6)

• Change “we propose a systematic review on” to “we present a systematic review of”.

=> Modified. (p. 6)

• I do not understand what the research question “until which frequency a species can perceive flicker?” means. Can this be rephrased please?

=> We indeed rephrased our hypotheses to better grasp the ecological matters we tried to answer: “This work focused on wild and domestic animals, excluding humans, and aimed at building a comprehensive knowledge base on animals’ CFF as well as answering the following questions: what evidence exists regarding animal CFF? What is the distribution of CFF between species? Which species are more at risk of being impacted by artificial lighting flicker?” (p. 6-7)

Materials and Methods

Page 10

• Change “which both measure the electrical response of the retina or brain to flickering light.” to “which measure the electrical response of the retina or brain to flickering light, respectively.”

=> Modified. (p. 10)

• Change “require an animal” to requires an animal”.

=> Modified. (p. 11)

Page 11

• I have concerns about how species were categorised based on trophic status. In the CFF database (file S10) species are coded on a four-point scale from 0-3. However, in the text the authors state that species were classified as being a primary consumer, omnivorous (this should be omnivore and predator). How do these three categories match onto the four-point scale of 0-3? Also, what constitutes a predator? Does this category include secondary and tertiary consumers? How were different species assigned into these ecological categories? Was this based on the literature, or intuition, or the authors own knowledge? Did the authors try to further separate predators into secondary and tertiary consumers? Having tried to classify large datasets of species into different trophic level I do appreciate that it is very difficult to classify such a wide range of animals, but I wonder if using such a simplistic scale means that important biological information is lost here. For example, all of the elasmobranchs within the database are classified as having the same trophic level (3). However, although they are all predators, they actually operate on different trophic levels in aquatic food webs. For example, rays and guitarfish feed on invertebrates and can be considered secondary consumers, while sharks such as scalloped hammerhead sharks, feed on rays and so are tertiary consumers. In addition, although not included in this study, there are other apex predator species of shark that occupy the roles of quaternary consumers, that will feed on both rays and scalloped hammerhead sharks.

=> Based on this comment, we extracted trophic guild data from available online databases, mainly FishBase (www.fishbase.org) and Animal Diversity Web (https://animaldiversity.org) as well as numerous other references to try to improve our trophic guild dataset (see CFF database for all references). We fit animals into 3 categories to limit problems of data overcategorisation in our model: ‘Herbivore’, ‘Omnivore’ or ‘Carnivore’.

=> We rephrased: “Each species was associated with its taxonomic class [48], its trophic guild (i.e. herbivore, omnivore, carnivore)…” (p. 11)

=> This new improved dataset changed the volume of CFF data that could be used in our model and we had to redo our analyses accordingly (see response to Editor)

• I also think the authors need to provide more information and clarity about how all of the species in the data based were classified in terms of body size. In a study dealing with so many different forms of animal life ranging across several orders of magnitude, how is it possible to divide them into large, average and small? For a species to be grouped into one of these three categories, did it have to be above or below a certain body size, for example? Also, what does the average category mean? Average body size in respect to what? All species in the animal kingdom, species included in this study? Perhaps the authors mean small, mid-size and large as opposed to small, average and large? More information is required. I also find it strange that whatever scheme the authors have used has resulted in, for example, all of the insects and birds being grouped in the same category (small), even though a house fly may be a few mm in size, weighing a fraction of a gram, whereas a Harris hawk can weigh 800-1000g and have a wingspan of over 1 metre. The system used also means that the Great horned owl, which is a very large bird with a wingspan of over 1 metre is classified as ‘small’, whereas a domestic cat, which weighs more than a great horned owl, but which is smaller in terms of body size/length, as classified as ‘average’. Another example is that a trout (30-60 cm, 05-3 kg), cuttlefish (30-40 cm, 2-4 kg) and sheep (50-100 kg) are all considered to be ‘average’. At the very least the authors need to better define and quantify their body size categories, and I think they should potentially consider have a broader range of categories.

=> Based on this comment, we extracted body mass data from available online databases, mainly FishBase (www.fishbase.org) and Animal Diversity Web (https://animaldiversity.org) as well as numerous other references to try to improve our body mass dataset (see CFF database for all references), notably on Insecta and Malacostraca. However, despite all our endeavours, very few body mass data were available for these two taxonomic classes. We then categorised animals with a body mass inferior to 1 g as ‘Very small’, between 1 g and 103 g ‘Small’, between 103 g and 104 g ‘Medium’, between 104 g and 105 g ‘Large’, and superior to 105 g ‘Very large’.

=> We rephrased and added: “Each species was associated with its taxonomic class [48], its trophic guild (i.e. herbivore, omnivore, carnivore) and a measure of its body size (i.e. very small, small, medium, large, very large). Categories were build thanks to data mainly provided by two available online trait databases [49,50]. Animals with a body mass inferior to 1 g were arbitrarily considered to be very small, between 1 g and 103 g small, between 103 g and 104 g medium, between 104 g and 105 g large, and superior to 105 g very large.” (p. 11)

=> This new improved dataset changed the volume of CFF data that could be used in our model and we had to redo our analyses accordingly (see answer to editor)

Results

Page 13

• Change “two-force choice procedures” to “two-alternative forced choice procedure”.

=> Modified. (p. 15)

• I do not understand the rationale for presenting the CFF values a for a species in the text, both on page 13 and throughout the manuscript. Why is the mean value plus/minus standard deviation or some other measure of variability enclosed within parentheses? For example, I would change “to (0.57 ± 0.08) Hz for the snail Lissachatina fulica B.” to “to 0.57 (± 0.08) Hz for the snail Lissachatina fulica B

=> Modified throughout the MS.

• Results in general. I think the results and analysis are fine as the manuscript stands, but if the authors end up revising their categorises for body size and trophic level, for example, then this will have an impact on the results and statistical analysis.

=> We modified the result section according to our new dataset and statistical analyses (see answer to Editor)

Discussion

Page 16.

• The results show that insects and birds have the highest CFF values. A logical next step for me at least would be to think that there is some correlation between needing a faster visual system and the sensory demands of flight. However, the authors do not seem to even mention this. Are flying animals more or less likely to be exposed to anthropogenic flickering light sources?

=> We indeed did not discuss this matter at first. We added a short sentence and cited Inger et al. (2014) and Boström et al. (2016) who talked about this matter: “The ability to fly, which many birds and insects have, could explain their high critical fusion frequencies since fast visual systems may be required to accomplish accurate manoeuvres and possibly avoid collisions [29,33].” (p. 19-20)

Page 17

• In the discussion of CFF in sharks the authors mention that McComb et al. hypothesized that variation in CFF between bonnethead, scalloped hammerhead and blacknose sharks could be because bonnethead and scalloped hammerhead sharks (which have relatively higher CFFs) live in shallow and bright reef habitats and feed on their fast-moving prey compared blacknose sharks, which usually forages in deeper and dimmer water environments. However, in table S10 the authors have classified bonnetheads as experiencing ‘high’ light exposure while both scalloped hammerheads and blacknose sharks are classified as having ‘low’ light exposure. It seems that the light level classifications assigned to these species contradicts the primary source of information. It also seems strange that the authors have highlighted these sharks as an example when their own coding of these specie sin their database appears to contradict the primary reference source.

=> We used Healy et al. (2013)’s nomenclature, “as the light levels of species that inhabit turbid waters are typically orders of magnitude lower than typical daylight levels where light levels are comparable to nocturnal light levels (Palmer & Grant 2010), we categorized these species as inhabiting low light level environments.”. As such, Healy et al. (2013) categorised the scalloped hammerhead as living in ‘Low’ light exposure levels because McComb et al. (2013) described them as living in turbid while shallow waters.

=> We thus decided to clarify our sentence and to only talk about clear water inhabiting species, namely the bonnethead and blacknose sharks: “Likewise, in clear water marine habitats, McComb et al. (2010) [30] related the higher temporal resolutions of the bonnethead sharks to their shallow and bright reef habitats and their fast-moving prey compared to the lower temporal resolution of blacknose sharks, which may forage in deeper and dimmer water environments.” (p. 20)

• Change “Our data showed that primary consumer” to “Our data showed that primary consumers”.

=> Based on new statistical analyses, this sentence was removed. (p. 20)

• Change “Such assumptions should, nevertheless, be studied” to “Such assumptions should be studied”.

=> Based on new statistical analyses, this sentence was removed. (p. 20)

Page 18

• Change “When the latter are still used for outdoor lighting” to “While the latter are still used for outdoor lighting”.

=> Modified. (p. 22)

• What does “Chatterjee et al., (2020) [61] brought the latest data” mean? What do the authors mean by ‘brought’ in this context?

=> We corrected this error thanks to the editor’s suggestion and rephrased as follows: “Chatterjee et al. (2020) [61] provided the latest data” (p. 22)

Page 19

• Change “For as crucial assessing an animal actual perception of flicker is” to “As crucial as assessing an animals actual flicker perception is”.

=> As pointed by the editor, this whole sentence was unclear. We rephrased it as follows: “Eventually, we would like to point out that animals may be subjected to the deleterious effects of flicker even though they cannot perceive it consciously. Indeed, Lu et al. (2012) [62] argued that a chromatic flicker at frequencies between 42.5 and 75 Hz, superior to the human CFF and therefore consciously unperceivable, was still able to entail their human subjects’ alerting and orienting attentional networks. Even though a first assessment of CFF seems essential, such results could challenge their wider use and justifies the need for a potential future systematic review on the specific matter of the impacts of flashing and flickering light on biodiversity.” (p. 23)

• Change “As it was first noted by Inger et al. (2014) [33] and even if more and more studies have been carried out lately, an important lack of data on animals’ CFF remains and is hampering our understanding of the impacts of anthropogenic lighting on biodiversity” to “As was first noted by Inger et al. (2014) [33], this study highlights how an overall lack of data on animals’ CFF remains and continues to hamper our understanding of the impacts of anthropogenic lighting on biodiversity.”

=> Modified. (p. 24)

Page 20

• Change “There, nevertheless, remains many recent studies using only one specimen or failing to report the use of a reference electrode for electroretinograms” to “Many recent studies have only used one specimen or have failed to report the use of a reference electrode for electroretinograms”.

=> Modified. (p. 25)

Page 22

• Change “outdoor lightings. Then, this result is necessarily provisional.” to “outdoor lighting. Therefore, the results presented here are necessarily provisional.”

=> Modified. (p. 27)

---

## [Editor Report · Decision Letter 1]

28 Nov 2022

PONE-D-22-13677R1A flashing light may not be that flashy: a systematic review on critical fusion frequenciesPLOS ONE

Dear Dr. Lafitte,

Thank you for submitting your manuscript to PLOS ONE. After careful consideration, we feel that it has merit but does not fully meet PLOS ONE’s publication criteria as it currently stands. Therefore, we invite you to submit a revised version of the manuscript that addresses the points raised during the review process.

The authors have responded to reviewer comments and made extensive revisions. The manuscript is improved and nearly acceptable. I suggest one additional revision. The formulation of research questions is an improvement. However, the first question, "what evidence exists regarding animal CFF?," does not seem to capture the authors' objectives. That question suggests that CFF might not exist or is somehow controversial. Perhaps the authors can delete this first question, elevate their second question to the first question and add a new second question that connects to the statistical analysis of differences among taxa (Table 1). For example: Are there differences in CFF between major taxonomic lineages? (or something to similar).

We look forward to receiving your revised manuscript.

Kind regards,

Christopher Nice, Ph.D.

Academic Editor

PLOS ONE
---

## [Author Response · Author response to Decision Letter 1]

5 Dec 2022

Editor: 

The authors have responded to reviewer comments and made extensive revisions. The manuscript is improved and nearly acceptable. I suggest one additional revision. The formulation of research questions is an improvement. However, the first question, "what evidence exists regarding animal CFF?," does not seem to capture the authors' objectives. That question suggests that CFF might not exist or is somehow controversial. Perhaps the authors can delete this first question, elevate their second question to the first question and add a new second question that connects to the statistical analysis of differences among taxa (Table 1). For example: Are there differences in CFF between major taxonomic lineages? (or something to similar).

=> We are thankful for your acknowledgement of our work in revising the MS. Based on your comment and because we acknowledge this first question might be misleading, we revised our review questions: “what is the distribution of CFF between species? Are there differences in how flicker is perceived between taxonomic classes? Which species are more at risk of being impacted by artificial lighting flicker?”

---

## [Editor Report · Decision Letter 2]

13 Dec 2022

A flashing light may not be that flashy: a systematic review on critical fusion frequencies

PONE-D-22-13677R2

Dear Dr. Lafitte,

We’re pleased to inform you that your manuscript has been judged scientifically suitable for publication and will be formally accepted for publication once it meets all outstanding technical requirements.

Kind regards,

Christopher Nice, Ph.D.

Academic Editor

PLOS ONE
---

## [Editor Report · Acceptance letter]

21 Dec 2022

PONE-D-22-13677R2 

A flashing light may not be that flashy: a systematic review on critical fusion frequencies 

Dear Dr. Lafitte:

I'm pleased to inform you that your manuscript has been deemed suitable for publication in PLOS ONE. Congratulations! Your manuscript is now with our production department. 

Kind regards, 

on behalf of

Dr. Christopher Nice 

Academic Editor

PLOS ONE